# Association between *XRCC3* p.Thr241Met polymorphism and risk of glioma: A systematic review and meta-analysis

Shing Cheng Tan[1]*, Teck Yew Low[1], Hafiz Muhammad Jafar Hussain[2], Mohamad Ayub Khan Sharzehan[1], Hilary Sito[1], Hamed Kord-Varkaneh[3], Md Asiful Islam[4]*

1 UKM Medical Molecular Biology Institute, Universiti Kebangsaan Malaysia, Kuala Lumpur, Malaysia, 2 Department of Molecular and Human Genetics, Baylor College of Medicine, Houston, TX, United States of America, 3 Department of Clinical Nutrition and Dietetics, Student Research Committee, Faculty of Nutrition and Food Technology, Shahid Beheshti University of Medical Sciences, Tehran, Iran, 4 Institute of Metabolism and Systems Research, University of Birmingham, Birmingham, United Kingdom

☉ These authors contributed equally to this work.
* sctan@ukm.edu.my, shingchengtan@gmail.com (SCT); ayoncx70@yahoo.com (MAI)

**Data Availability Statement:** All relevant data are within the paper and its Supporting Information files.

## Abstract

### Background

The *XRCC3* p.Thr241Met (rs861539) polymorphism has been extensively studied for its association with glioma risk, but results remain conflicting. Therefore, we performed a systematic review and meta-analysis to resolve this inconsistency.

### Methods

Studies published up to June 10, 2022, were searched in PubMed, Web of Science, Scopus, VIP, Wanfang, and China National Knowledge Infrastructure databases and screened for eligibility. Then, the combined odds ratio (OR) of the included studies was estimated based on five genetic models, i.e., homozygous (Met/Met vs. Thr/Thr), heterozygous (Thr/Met vs. Thr/Thr), dominant (Thr/Met + Met/Met vs. Thr/Thr), recessive (Met/Met vs. Thr/Thr + Thr/Met) and allele (Met vs. Thr). The study protocol was preregistered at PROSPERO (registration number: CRD42021235704).

### Results

Overall, our meta-analysis of 14 eligible studies involving 12,905 subjects showed that the p.Thr241Met polymorphism was significantly associated with increased glioma risk in both homozygous and recessive models (homozygous, OR = 1.381, 95% CI = 1.081–1.764, P = 0.010; recessive, OR = 1.305, 95% CI = 1.140–1.493, P<0.001). Subgroup analyses by ethnicity also revealed a statistically significant association under the two aforementioned genetic models, but only in the Asian population and not in Caucasians (P>0.05).

**Funding:** Research in SCT's laboratory is supported by the Research University Grant of Universiti Kebangsaan Malaysia (No. GUP-2020-076), the Fundamental Research Grant Scheme of the Ministry of Higher Education, Malaysia (No. FRGS/1/2019/SKK08/UKM/02/9), and the Higher Institution Center of Excellence (HICoE) grant of the Ministry of Higher Education, Malaysia (No. AKU49). The funders had no role in study design, data collection and analysis, decision to publish, or preparation of this manuscript.

**Competing interests:** The authors declare no conflicts of interest.

## Conclusion

We demonstrated that the *XRCC3* p.Thr241Met polymorphism is associated with an increased risk of glioma only in the homozygous and recessive models.

## Introduction

Gliomas refer to a type of tumor that originates from the glial cells located in the brain or spine. This type of cancer accounts for ~30% of tumors of the brain and central nervous system and ~80% of total cases of malignant brain tumors [1, 2]. The tumor has a five-year survival rate of approximately 60%. However, this rate can be improved to as high as 74% if adequate surveillance by biopsy is performed in conjunction with early resection for low-grade gliomas [3, 4]. Depending on their aggressiveness, gliomas can be classified into four different grades, namely grade I, II, III, and IV. While grade I and II gliomas (low-grade gliomas) are comparatively benign and grow slowly, grade III (malignant glioma) and IV (glioblastoma multiforme or glioblastoma) gliomas are very aggressive and grow rapidly. Notably, glioblastoma is the most common and deadly form of primary malignant glioma in adults. It accounts for ~70% of gliomas and has a median survival of only 12–14 months [2, 5].

Although gliomas are a heterogeneous disease with comparatively unclear etiology, risk factors for gliomas are thought to consist of genetic predisposition and environmental influences [2, 3]. Genetic disorders such as type 1 and type 2 neurofibromatosis and tuberous sclerosis complex [6], as well as obesity and body height [7], have been shown to predispose to gliomas. In addition, several genome-wide association studies (GWAS) have reported the involvement of multiple genetic polymorphisms in mediating glioma risk [8, 9]. On the other hand, environmental factors such as ionizing radiation, ultraviolet radiation, cytomegalovirus infection, environmental carcinogens, and diet have been associated with an increased risk of developing glioma, although only the first two risk factors have been clearly demonstrated [3, 10–13].

Among a variety of genetic factors, germline polymorphisms have been widely associated with cancer risk [14, 15]. In particular, polymorphisms of DNA repair genes such as *ERCC1* [16], *ERCC2 (XPD)* [17], *XRCC1* [18], *XRCC3* [18], *XRCC4* [19], *XRCC5*, *XRCC6* [20], *XRCC7* [21], *MGMT* [22], *CHAF1* [23], *LIG4* [19], and *GLTSCR1* [17] have been associated with glioma risk [24, 25]. As genetic material, DNA is constantly exposed to endogenous and exogenous attacks that can manifest themselves in cellular metabolic processes and genotoxic or clastogenic stresses, including ionizing radiation and ultraviolet radiation mentioned earlier. These attacks can cause cross-links between DNA and proteins, oxidative damage to DNA, and single- and double-strand breaks in DNA chains [26]. However, this damage, which can affect the integrity of the genome, is continuously and effectively repaired by various inherent DNA repair pathways. These cellular DNA repair pathways can function in several ways and rely on the mechanisms of homologous recombination repair (HRR), nonhomologous end joining, base excision repair, and nucleotide excision repair. However, common polymorphisms in these DNA repair genes can impair DNA repair, leading to a higher risk of developing gliomas and other forms of malignancy [18–22].

This meta-analysis focuses on the *XRCC3* (X-ray repair cross complementing 3) gene, which encodes a RecA/Rad51-related protein involved in HRR. Of the polymorphisms in *XRCC3*, only p.Thr241Met (rs861539) has been frequently studied with respect to its association with glioma risk. This polymorphism is located in exon 7 of the gene and involves a C-to-T transition at codon 241 [27]. Unlike other polymorphisms in *XRCC3*, the nonsynonymous

p.Thr241Met polymorphism causes the replacement of a polar amino acid (threonine) with a nonpolar one (methionine), which can significantly affect the functionality of the protein product. For this reason, this polymorphism has been studied in numerous malignancies, including bladder cancer, head and neck cancer, ovarian cancer, and colorectal cancer, and significant associations have been demonstrated in some cases [28–32]. However, in gliomas, the association between this polymorphism and susceptibility to this cancer remains unclear [18, 21, 24, 33–44]. To clarify this, we performed a meta-analysis to further investigate the association between the *XRCC3* p.Thr241Met polymorphism and glioma risk.

## Materials and methods

### Literature search

We comprehensively searched PubMed, Web of Science, Scopus, VIP, Wanfang, and China National Knowledge Infrastructure databases for relevant studies, including grey literature, published through June 10, 2022. The terms used in these searches were: (brain tumor OR glioma OR glioblastoma OR astrocytoma OR oligodendroglioma OR GBM OR glioblastoma multiforme) AND (XRCC3 OR X-Ray repair cross complementing 3 OR DNA repair gene) AND (polymorphism OR mutation OR variant OR variation). No language restriction was applied. We then included studies that met our prespecified inclusion criteria, i.e., (i) examined the association between the *XRCC3* p.Thr241Met polymorphism and glioma risk, (ii) were observational studies, such as case-control studies, and (iii) reported genotype and/or allele frequencies or adequate information to infer these frequencies. We focused only on observational studies because other types of studies, such as randomized controlled trials, are not ethically acceptable in the context of genetic studies. In contrast, we excluded a study if (i) it was not an original study, (ii) the research was performed on cell lines, animal models, or other non-human subjects, and (iii) the study was a duplication of other publications. We also pre-registered our study protocol with PROSPERO (registration number: CRD42021235704).

### Extraction of data and quality appraisal

Two investigators independently extracted the following information from the eligible studies: (i) first author's name, (ii) publication year, (iii) ethnicity, (iv) source of controls (hospital-based vs. population-based), (v) country where the study was conducted, (vi) number of subjects per group, (vii) frequency of polymorphic genotypes and alleles, and (viii) Hardy-Weinberg equilibrium (HWE) of genotype distribution in controls. If data on HWE were missing in any of the studies, a goodness-of-fit test was performed. If data were missing, the corresponding author of the included studies was contacted by email. The quality of the studies was then assessed using a modified version of the Newcastle-Ottawa Scale [45], which assessed (i) whether the definition of cases was appropriate, (ii) whether cases were representative of populations, (iii) whether controls were recruited from the community and the genotype distribution conformed to HWE, (iv) whether controls were defined as having no history of disease, (v) whether cases and controls were comparable in terms of ethnic homogeneity, (vi) whether there was no evidence of population stratification, (vii) whether quality control procedures and blinding were used in genotyping, (viii) whether cases and controls were genotyped using the same method, and (ix) whether the genotyping call rate exceeded 99% (S1 File). If the answer to each of the above criteria was "yes", the study received one star for each criterion. A study had to receive at least 5 stars to be considered of high methodological quality.

## Meta-analysis

STATA (ver. 16.0) was used to pool data from included studies. P < 0.05 was considered statistically significant, unless otherwise stated. The association between *XRCC3* p.Thr241Met polymorphism and glioma risk was assessed using five genetic models: homozygous (Met/Met vs. Thr/Thr), heterozygous (Thr/Met vs. Thr/Thr), dominant (Thr/Met + Met/Met vs. Thr/Thr), recessive (Met/Met vs. Thr/Thr + Thr/Met), and allele (Met vs. Thr). Examination of multiple genetic models was required because genetic association studies do not adopt a specific model [46]. Subsequently, Cochran's Q and $I^2$ tests were used to measure the heterogeneity of the studies, which was considered high when P<0.1 or $I^2$ >50% [47]. In this case, a random-effects method was used to derive the odds ratio (OR). On the other hand, when heterogeneity was low, a fixed-effects method was used. The Z test was then used to measure the significance of the pooled OR, and forest plots were created to visually display the results. Subgroup analyses were then conducted by several variables, including ethnicity, source of control, and HWE status, as these variables have long been known to affect genetic associations [48–50]. To examine whether the results were primarily influenced by one of the studies, we also performed a sensitivity analysis using the leave-one-out method. Publication bias was assessed using Begg's test, and Egger's test, and visual inspection of funnel plots.

## Results

### Study selection

We first identified 2,052 entries from the search databases (PubMed, N = 1,127; Scopus, N = 46; WoS, N = 838; CNKI, N = 9; Wanfang, N = 19; VIP, N = 13). We then removed 488 duplicate entries and subjected the remaining 1,564 articles to title and abstract screening. This process identified only 27 articles as potentially relevant. We then screened these 27 articles for eligibility by reviewing the full texts. This resulted in a further exclusion of 13 articles, so that finally only 14 articles describing 14 relevant studies were considered for quantitative data synthesis. The flow diagram of the study selection is shown in Fig 1.

### Characteristics of the studies

The final 14 eligible studies included a total of 12,905 study participants, including 5,852 cases and 7,053 controls (Table 1). Eight (8) of these 14 studies were conducted in China, with subjects from the Asian population [28, 33, 37, 39, 40, 42–44]; the other seven studies were predominantly Caucasian–with three studies from the United States [21, 24, 36] and one each from Spain [41] and Brazil [34], and one study from multiple countries [18]. In addition, 11 studies used hospital-based controls [21, 24, 33, 36, 37, 39–44], whereas the remaining 3 studies opted for population-based controls [18, 34, 36]. In addition, all studies except Liu et al. [36] reported the frequencies of the three genotypes (Thr/Thr, Thr/Met, and Met/Met) separately. Instead, Liu et al. 2009 [36] combined the homozygous wild type (Thr/Thr) and heterozygous genotypes (Thr/Met). Moreover, of all 14 studies, the distribution of control genotypes did not deviate significantly from HWE in only five studies (P>0.05) [21, 33, 37, 39, 41]. Nevertheless, all 14 studies were of high quality (S1 Table).

### Quantitative data synthesis

The combined results of the 14 studies are shown in Table 2, and the corresponding forest plots are shown in Fig 2. High statistical heterogeneity was detected under in all genetic models; therefore, a random-effects model was used in the data synthesis. We detected a statistically significant association between the *XRCC3* p.Thr241Met polymorphism and increased glioma

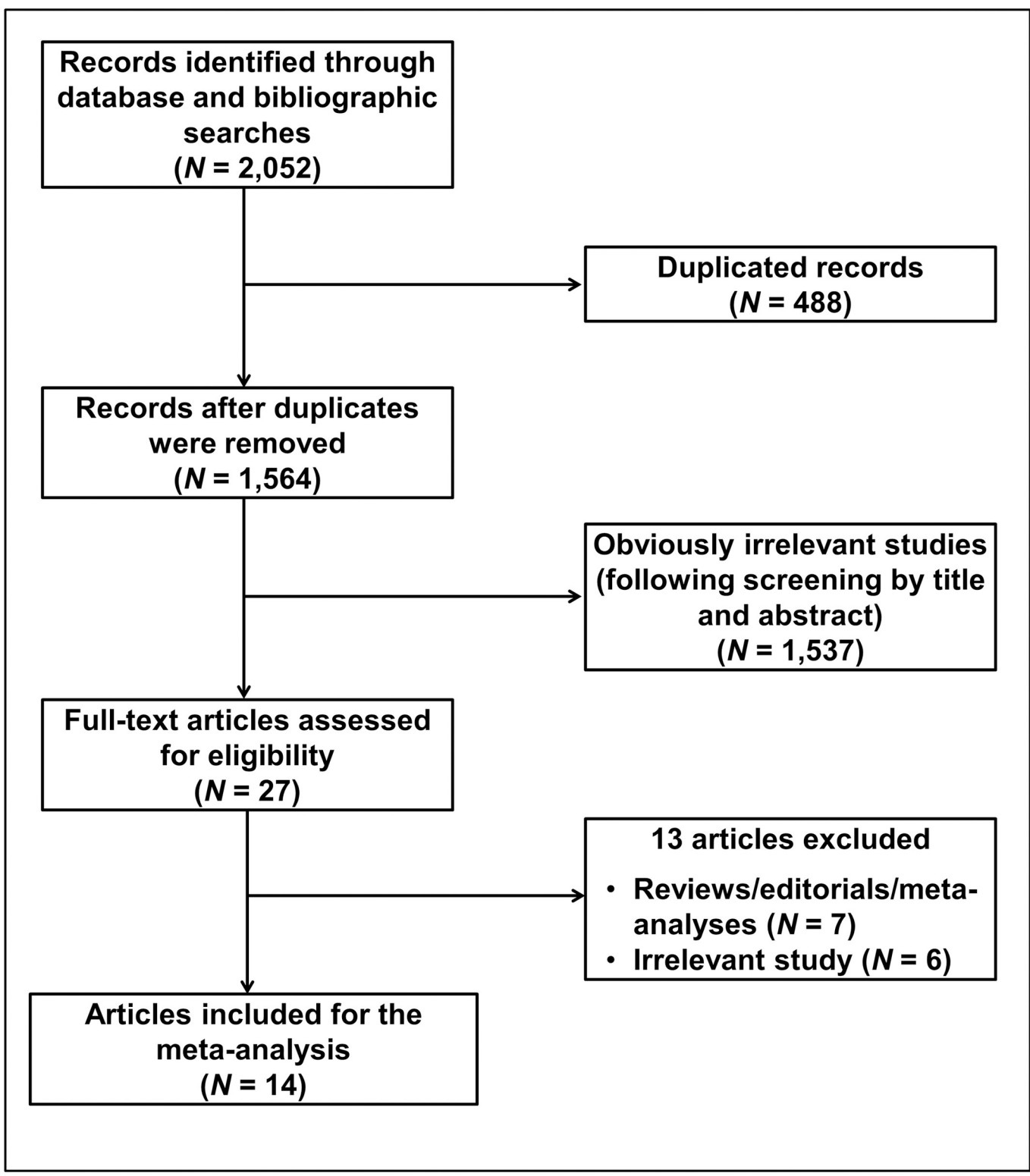

**Fig 1. Flow diagram of study selection.**

**Table 1. Characteristics of the included studies.**

| Study ID [reference] | Country | Ethnicity | Source of control* | Cases | | | Controls | | | HWE P-value (controls) |
|---|---|---|---|---|---|---|---|---|---|---|
| | | | | Thr/Thr | Thr/Met | Met/Met | Thr/Thr | Thr/Met | Met/Met | |
| Huang 2015 [39] | China | Asian | HB | 310 | 72 | 7 | 239 | 111 | 8 | 0.237 |
| Gao 2014 [40] | China | Asian | HB | 158 | 146 | 22 | 202 | 159 | 15 | 0.016 |
| Rodriguez-Hernandez 2014 [41] | Spain | Caucasian | HB | 43 | 56 | 16 | 87 | 92 | 21 | 0.646 |
| Xu 2014 [42] | China | Asian | HB | 472 | 343 | 71 | 485 | 356 | 45 | 0.047 |
| Luo 2013 [43] | China | Asian | HB | 145 | 131 | 21 | 229 | 168 | 17 | 0.042 |
| Pan 2013 [44] | China | Asian | HB | 217 | 198 | 28 | 234 | 200 | 9 | <0.001 |
| Zhao 2013 [33] | China | Asian | HB | 336 | 47 | 1 | 340 | 41 | 3 | 0.165 |
| Custódio 2012 [34] | Brazil | Caucasian | PB | 53 | 18 | 9 | 86 | 9 | 5 | <0.001 |
| Liu 2012 [35] | China | Asian | HB | 223 | 154 | 66 | 254 | 147 | 42 | 0.003 |
| Rajaraman 2010 [24] | USA | Caucasian | HB | 135 | 162 | 53 | 185 | 208 | 86 | 0.042 |
| Liu 2009 [36] | USA | Caucasian | PB | 308 | | 61 | 315 | | 45 | - |
| Zhou 2009 [37] | China | Asian | HB | 677 | 80 | 3 | 629 | 75 | 4 | 0.286 |
| Kiuru 2008 [18] | Multiple | Caucasian | PB | 288 | 319 | 94 | 630 | 761 | 169 | 0.006 |
| Wang 2004 [21] | USA | Caucasian | HB | 134 | 138 | 37 | 147 | 147 | 48 | 0.254 |

* HB, hospital-based; PB, population-based

risk when the homozygous model and the recessive model were applied (homozygous, OR = 1.381, 95% CI = 1.081–1.764, P = 0.010; recessive, OR = 1.305, 95% CI = 1.140–1.493, P<0.001). However, no significant association was found in the heterozygous (OR = 1.040, 95% CI = 0.908–1.191, P = 0.574), dominant (OR = 1.082, 95% CI = 0.939–1.247, P = 0.277), and allele (OR = 1.097, 95% CI = 0.972–1.237, P = 0.133) models. Sensitivity analysis revealed that the pooled OR was not significantly affected by any of the studies (S1 Fig).

## Subgroup analysis

In the ethnicity-based subgroup analysis, we again found a statistically significant association between *XRCC3* p.Thr241Met and increased glioma risk in the homozygous (OR = 1.704, 95% CI = 1.358–2.317, P<0.001) and recessive models (OR = 1.305, 95% CI = 1.140–1.493, P<0.001) in the Asians (Table 2). In contrast, no significant association was observed in Caucasians in all genetic models (P>0.05). Subgroup analysis by source of controls revealed that the association in hospital-based controls was significant only in the homozygous model (OR = 1.364, 95% CI = 1.015–1.832, P = 0.040), whereas in population-based controls, the polymorphism was associated with increased glioma risk only in the recessive model (OR = 1.338, 95% CI = 1.071–1.670, P = 0.010). Interestingly, subgroup analysis by HWE status revealed no significant association for studies whose genotype distribution conformed to the equilibrium. For studies in which the genotype distribution deviated significantly from HWE, a significant association was found in the homozygous (OR = 1.588, 95% CI = 1.200–2.102, P = 0.001), dominant (OR = 1.124, 95% CI = 1.027–1.229, P = 0.011), recessive (OR = 1.518, 95% CI = 1.164–1.981, P = 0.002), and allele (OR = 1.185, 95% CI = 1.053–1.333, P = 0.005) models.

## Publication bias

We did not detect any noticeable asymmetry in any of the funnel plots (Fig 3), indicating that there was no publication bias. This finding was also supported by the results of Begg's and Egger's tests (homozygous, Begg's P = 1.000, Egger's P = 0.811; heterozygous, Begg's P = 0.222,

**Table 2. Summary of the association between *XRCC3* p.Thr241Met polymorphism and glioma risk.**

| Comparison model | | No. of studies | No. of cases | No. of controls | Effect model | OR (95% CI) | P |
|---|---|---|---|---|---|---|---|
| Homozygous model | | | | | | | |
| | Overall | 13 | 3,619 | 4,219 | Random | 1.381 (1.081–1.764) | 0.010 |
| | Asian | 8 | 2,757 | 2,755 | Fixed | 1.704 (1.358–2.137) | <0.001 |
| | Caucasian | 5 | 862 | 1,464 | Fixed | 1.094 (0.895–1.338) | 0.378 |
| | Hospital-based controls | 11 | 3,175 | 3,329 | Random | 1.364 (1.015–1.832) | 0.040 |
| | Population-based controls | 2 | 444 | 890 | Random | 1.570 (0.720–3.425) | 0.257 |
| | Conform to HWE | 5 | 1,564 | 1,526 | Fixed | 0.915 (0.636–1.317) | 0.634 |
| | Deviate from HWE | 8 | 2,055 | 2,693 | Random | 1.588 (1.200–2.102) | 0.001 |
| Heterozygous model | | | | | | | |
| | Overall | 13 | 5,055 | 6,221 | Random | 1.040 (0.908–1.191) | 0.574 |
| | Asian | 8 | 3,709 | 3,869 | Random | 1.009 (0.841–1.210) | 0.925 |
| | Caucasian | 5 | 1,346 | 2,352 | Random | 1.102 (0.872–1.394) | 0.415 |
| | Hospital-based controls | 11 | 4,377 | 4,735 | Random | 1.028 (0.896–1.180) | 0.694 |
| | Population-based controls | 2 | 678 | 1,486 | Random | 1.602 (0.468–5.480) | 0.453 |
| | Conform to HWE | 5 | 1,893 | 1,908 | Random | 0.923 (0.663–1.284) | 0.633 |
| | Deviate from HWE | 8 | 3,162 | 4,313 | Fixed | 1.060 (0.964–1.164) | 0.228 |
| Dominant model | | | | | | | |
| | Overall | 13 | 5,483 | 6,693 | Random | 1.082 (0.939–1.247) | 0.277 |
| | Asian | 8 | 3,928 | 4,012 | Random | 1.056 (0.872–1.279) | 0.578 |
| | Caucasian | 5 | 1,555 | 2,681 | Random | 1.129 (0.888–1.435) | 0.321 |
| | Hospital-based controls | 11 | 4,702 | 5,033 | Random | 1.058 (0.915–1.224) | 0.446 |
| | Population-based controls | 2 | 781 | 1,660 | Random | 1.649 (0.527–5.162) | 0.390 |
| | Conform to HWE | 5 | 1,957 | 1,992 | Random | 0.914 (0.667–1.253) | 0.577 |
| | Deviate from HWE | 8 | 3,526 | 4,701 | Fixed | 1.124 (1.027–1.229) | 0.011 |
| Recessive model | | | | | | | |
| | Overall | 14 | 5,852 | 7,053 | Fixed | 1.305 (1.140–1.493) | <0.001 |
| | Asian | 8 | 3,928 | 4,012 | Fixed | 1.655 (1.327–2.065) | <0.001 |
| | Caucasian | 6 | 1,924 | 3,041 | Fixed | 1.132 (0.954–1.343) | 0.155 |
| | Hospital-based controls | 11 | 4,702 | 5,033 | Random | 1.316 (0.992–1.748) | 0.057 |
| | Population-based controls | 3 | 1,150 | 2,020 | Fixed | 1.338 (1.071–1.670) | 0.010 |
| | Conform to HWE | 5 | 1,957 | 1,992 | Fixed | 0.911 (0.645–1.286) | 0.594 |
| | Deviate from HWE | 8 | 3,526 | 4,701 | Random | 1.518 (1.164–1.981) | 0.002 |
| Allele model | | | | | | | |
| | Overall | 13 | 5,483 | 6,693 | Random | 1.097 (0.972–1.237) | 0.133 |
| | Asian | 8 | 3,928 | 4,012 | Random | 1.087 (0.923–1.281) | 0.316 |
| | Caucasian | 5 | 1,555 | 2,681 | Random | 1.106 (0.910–1.344) | 0.311 |
| | Hospital-based controls | 11 | 4,702 | 5,033 | Random | 1.069 (0.942–1.214) | 0.302 |
| | Population-based controls | 2 | 781 | 1,660 | Random | 1.621 (0.625–4.203) | 0.320 |
| | Conform to HWE | 5 | 1,957 | 1,992 | Random | 0.915 (0.714–1.174) | 0.486 |
| | Deviate from HWE | 8 | 3,526 | 4,701 | Random | 1.185 (1.053–1.333) | 0.005 |

Egger's P = 0.185; dominant, Begg's P = 0.542, Egger's P = 0.340; recessive, Begg's P = 0.784, Egger's P = 0.857; allele, Begg's P = 0.542, Egger's P = 0.643).

## Discussion

The XRCC3 protein belongs to the RecA/Rad51-related family of proteins involved in HRR. This protein is encoded by a ~18kb gene, *XRCC3*, located on chromosome 14q32.33 [51]. The

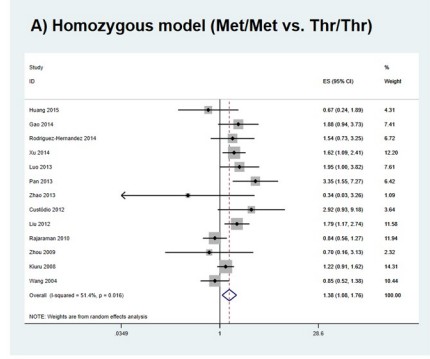 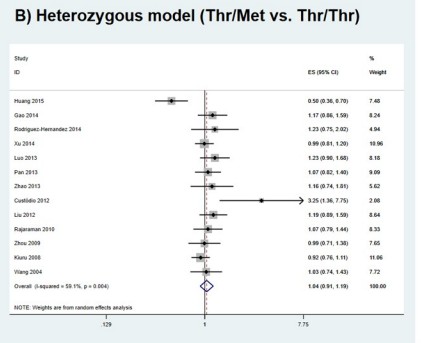 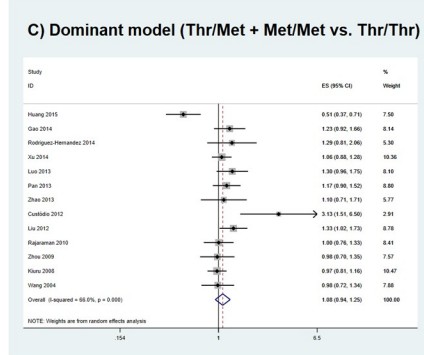

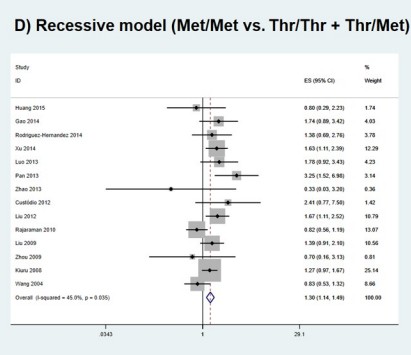 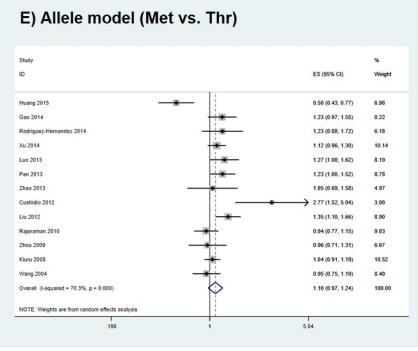

**Fig 2. Forest plots of the association between *XRCC3* p.Thr241Met polymorphism and glioma risk.**

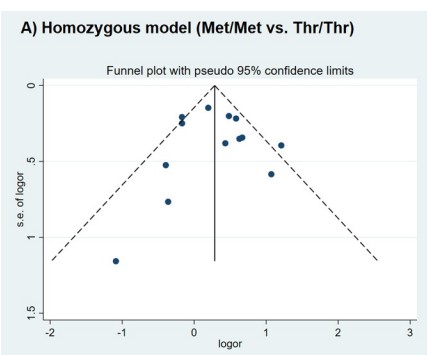 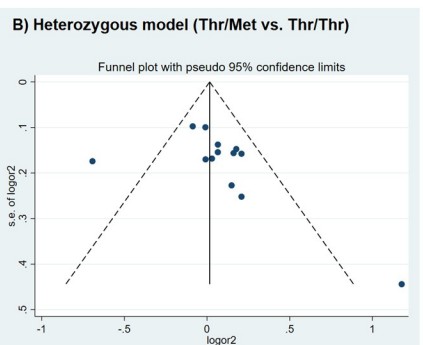 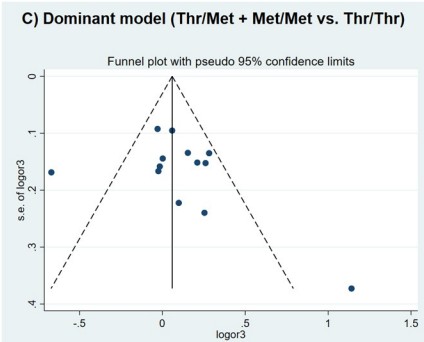

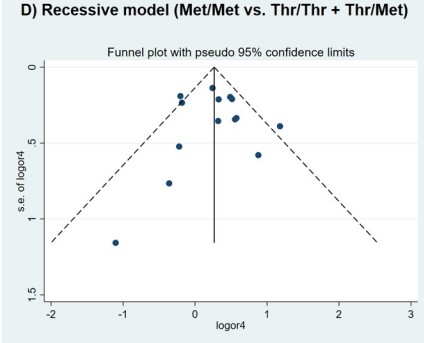 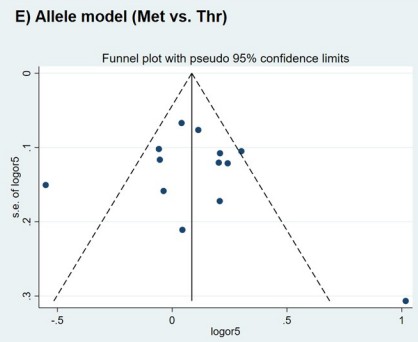

**Fig 3.**

gene product is a ~38kDa protein that belongs to the five paralogs of RAD51, i.e., XRCC2, XRCC3, RAD51B, RAD51C, and RAD51D, all of which share 20–30% structural similarity with RAD51 [52]. These paralogs typically function by forming complex assemblies that can facilitate RAD51-mediated activities to repair DNA double-strand breaks via HRR. Therefore, depletion of any of these paralogs often leads to a reduction in the DNA damage response [53]. Two important protein complexes formed by these paralogs are (i) the BCDX2 complex, which includes XRCC2, RAD51B, RAD51C, and RAD51D, and (ii) the CX3 complex, which includes XRCC3 as well as RAD51C [54, 55]. These two complexes have been reported to be involved in two distinct steps of HRR: the BCDX2 complex contributes to the stabilization of the nucleoprotein filament of RAD51, whereas the CX3 complex acts after the recruitment of RAD51 to the damage sites [53]. Although the exact biochemical mechanisms of the CX3 complex have remained unclear, the human CX3 complex has been shown to promote RAD51 nucleofilament remodeling and stability, as well as strand invasion, whereas its other proposed role is to help recruit specialized factors to catalyze fork restoration through branch migration or controlled fork processing [56].

Reduced levels of XRCC3 protein have been shown to lead to increased rates of chromosome segregation errors, aneuploidy, and other chromosome aberrations [57]. Notably, cells deficient in XRCC3 have been shown to have lower HRR and hypersensitivity to cross-linking agents, genotoxic alkylating agents, UV radiation, and ionizing radiation [58, 59]. Similarly, common polymorphisms in this gene have been shown to lead to a derailed DNA damage response that may increase the likelihood of tumor development [60]. It is also known that *XRCC3* polymorphisms play an important role in the treatment of gliomas. In current practice, temozolomide is the chemotherapeutic agent of choice for high-grade gliomas [61]. Recently, the *XRCC3* polymorphism was found to contribute to temozolomide resistance in glioblastoma cells by mediating the repair of DNA double-strand breaks [62]. However, this may not be the whole story about the contribution of XRCC3 to temozolomide resistance. XRCC3 has also been found to contribute to mitochondrial biogenesis by promoting mitochondrial DNA integrity [63]. Mitochondria are known to enhance temozolomide chemoresistance, suggesting a role for XRCC3 in the treatment of gliomas [64, 65].

Given the important role of XRCC3 in HRR, its abundance and 3-D conformation require well-tuned regulation. As both protein structure and expression can be affected by genetic polymorphisms, a number of GWAS have investigated the association between *XRCC3* polymorphisms and risk for cancers, including gliomas [33, 37, 39, 66–68]. Among the polymorphisms in *XRCC3*, we focused on p.Thr241Met, an exonic polymorphism involving a C-to-T transition at codon 241 in exon 7 [27]. This transition replaces threonine (Thr) with methionine (Met) and may affect XRCC3 functions, activities, and interactions [69]. We chose to perform a meta-analysis of this polymorphism because p.Thr241Met has been extensively studied with respect to its association with gliomas, with often conflicting results. For example, whereas Custódio et al. [34] found that the Met allele of p.Thr241Met was significantly associated with a 3-fold increase in glioma risk, Huang et al. [39] found that the same allele was associated with a 1.6-fold decrease in glioma risk. On the other hand, Rodriguez-Hernandez et al. [41] showed that there was no significant association between the polymorphism and glioma risk. These inconsistencies could be due to numerous factors, such as the ethnicity of the study population, and the sample size (and thus statistical power) of the studies [70, 71]. A well-conducted meta-analysis is essential to address these inconsistencies. However, the most recent meta-analyses on this topic were conducted many years ago [72–77] and did not include several eligible studies. The lack of meticulous data extraction in meta-analysis can lead to misleading results [78]. Therefore, in this work, we meticulously performed a systematic review and meta-analysis to investigate the association between the *XRCC3* p.Thr241Met polymorphism and glioma risk.

Our results showed that the p.Thr241Met polymorphism was associated with increased glioma risk only in the homozygous and recessive models, but not in the heterozygous, dominant, and allele models. Within each model, we additionally clustered our analysis by different subgroups, i.e., ethnicity, source of controls, and HWE status. Our analysis in the Asian subgroup again showed that only the homozygous and recessive models were associated with increased glioma risk, whereas in the Caucasian subgroup, no significance was found in any of the genetic models. In addition, studies using hospital- and population-based controls appeared to result in different risk profiles, and studies in which the genotypic distribution of controls deviated from HWE showed more significant associations. It should be noted that the number of participants in each subgroup was much smaller than in the overall analysis. Therefore, there is a possibility that the subgroup analyses were not sufficiently powered to detect possible associations [79].

The greatest strength of our meta-analysis lies in the extensive collection of genetic association studies on the *XRCC3* p.Thr241Met polymorphism performed on a large population. Moreover, the selected parameter studied, i.e., the p.Thr241Met polymorphism, is a well-established genetic variation that can be accurately genotyped using contemporary genotyping technologies. Therefore, the genotyping results obtained in different studies are reliable and well comparable with each other. Nevertheless, there are some limitations to this study as well. First, the influence of gene-gene and gene-environment interactions was not investigated because information on this aspect was lacking in the included studies. In addition, the study population, especially that of Asian populations, was mainly from China and therefore not very diverse. Furthermore, our meta-analysis examined only one polymorphism, whereas analysis of multiple genetic polymorphisms in the same gene might provide a more complete picture of the role of *XRCC3* in mediating glioma risk.

## Conclusions

In conclusion, our results show that there is a significant association between the *XRCC3* p.Thr241Met polymorphism and increased glioma risk, but only when the homozygous and recessive models were adopted. When divided into subgroups by ethnicity, similar results were observed only in the Asian population. In addition, no obvious trend was observed when data were stratified by source of controls and HWE status, presumably because of low study power. Therefore, further genetic association studies are needed to provide a more accurate assessment of the association between the polymorphism and glioma risk.

## Supporting information

**S1 Table. Quality assessment of the included studies.**
(DOCX)

**S1 Fig. Sensitivity analysis of the meta-analysis.**
(ZIP)

**S1 File. Modified Newcastle-Ottawa scale for meta-analysis of genetic association studies.**
(DOCX)

## Acknowledgments

The authors thank Academic Proofreading (https://www.academicproofreading.uk) for English language editing.

## Author Contributions

**Conceptualization:** Shing Cheng Tan.

**Data curation:** Shing Cheng Tan, Mohamad Ayub Khan Sharzehan, Hilary Sito, Hamed Kord-Varkaneh.

**Formal analysis:** Shing Cheng Tan.

**Funding acquisition:** Shing Cheng Tan.

**Investigation:** Shing Cheng Tan, Teck Yew Low, Hafiz Muhammad Jafar Hussain, Mohamad Ayub Khan Sharzehan, Hilary Sito.

**Methodology:** Shing Cheng Tan, Mohamad Ayub Khan Sharzehan, Hilary Sito, Hamed Kord-Varkaneh.

**Software:** Hamed Kord-Varkaneh.

**Supervision:** Hafiz Muhammad Jafar Hussain.

**Validation:** Teck Yew Low, Mohamad Ayub Khan Sharzehan, Hilary Sito.

**Visualization:** Shing Cheng Tan.

**Writing – original draft:** Teck Yew Low.

**Writing – review & editing:** Shing Cheng Tan, Md Asiful Islam.

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
