## [Decision Letter · Decision Letter 0]

19 May 2022

PONE-D-21-33188

Association between XRCC3 p.Thr241Met polymorphism and risk of glioma: a systematic review and meta-analysis

PLOS ONE

Dear Dr. Tan,

Thank you for submitting your manuscript to PLOS ONE. After careful consideration, we feel that it has merit but does not fully meet PLOS ONE’s publication criteria as it currently stands. Therefore, we invite you to submit a revised version of the manuscript that addresses the points raised during the review process.

ACADEMIC EDITOR: Please revise the manuscript per the following comments.

We look forward to receiving your revised manuscript.

Kind regards,

Farzad Taghizadeh-Hesary

Academic Editor

PLOS ONE

**Journal requirements:**

“Research in SCT’s laboratory is supported by the Research University Grant of Universiti Kebangsaan Malaysia (No. GUP-2020-076) and the Fundamental Research Grant Scheme of the Ministry of Higher Education, Malaysia (No. FRGS/1/2019/SKK08/UKM/02/9). The funders had no role in study design, data collection and analysis, decision to publish, or preparation of this manuscript.”

“Research in SCT’s laboratory is supported by the Research University Grant of Universiti Kebangsaan Malaysia (No. GUP-2020-076) and the Fundamental Research Grant Scheme of the Ministry of Higher Education, Malaysia (No. FRGS/1/2019/SKK08/UKM/02/9). The funders had no role in study design, data collection and analysis, decision to publish, or preparation of this manuscript.”

“The authors declare no conflicts of interest.”

**Additional Editor Comments:**

Academic Editor:

1. Please follow the PLOS ONE guideline for preparing the manuscript.

2. Please revise the manuscript per the Reviewers' comments.

3. It is recommended the authors mention the importance of XRCC3 polymorphism in glioma treatment. In the current practice, temozolomide is the choice chemotherapeutic for high-grade gliomas (https://www.ncbi.nlm.nih.gov/pmc/articles/PMC8651479/). Recent evidence, has noted that XRCC3 polymorphism contributes to temozolomide resistance of glioblastoma cells by mediating DNA double-strand break repair (https://pubmed.ncbi.nlm.nih.gov/29574277/). However, this may not be the whole story of XRCC3 contribution to the temozolomide resistance. Recent evidence has noted that XRCC3 contributes to mitochondrial biogenesis by facilitating the mitochondrial DNA integrity (https://pubmed.ncbi.nlm.nih.gov/29158291/). Besides, it has been shown that mitochondria improves temozolomide chemoresistance (https://www.preprints.org/manuscript/202201.0171/v2, AND https://pubmed.ncbi.nlm.nih.gov/20870728/). It is recommended the authors mention this crucial issue in the Discussion section and cite all the noted articles.

Reviewer #1:

This study represents the largest to date meta-analysis to look at the association between the XRCC3 p.Thr241Met (rs861539) polymorphism and Glioma. It is a well-designed study, which has employed appropriate statistical techniques, and its findings are interesting albeit relatively minor advance to our understanding of glioma genetic risk.

1. Major comment:

The authors state that the last meta study to look at rs861539 and glioma risk was over a decade ago is not true. Qi et al 2017 and Feng et al 2014 both performed similar meta-analyses but did not find there to be an association. These more recent and very similar papers should have been referenced and the authors findings discussed within the context of this early work.

2. Language:

Unfortunately, language quality in many parts was poor, with many sentences consisting of peculiar or inappropriate wording. Thus, I feel it is very important the authors re-write the manuscript with assistance of a copy editor.

Reviewer #2:

1. Materials and Methods - Literature search (page 8, line 111): It may be not clear to readers why the authors consider the case-control type of studies only rather than others.

2. Materials and Methods - Literature search (page 8, line 115): It is vague for the exclusion "(iii) they were duplicates(s)."

3. Materials and Methods - Extraction of data and quality appraisal (page 9, line 126): I would encourage the authors to describe more in detail about the methods and procedures of quality appraisal.

4. Materials and Methods - Meta-analysis (page 9, line 133): Please highlight and broaden the foundations and considerations of the five genetic models here or elsewhere appropriate.

5. Materials and Methods - Meta-analysis (page 9, line 137): I would suggest provide a reference for the statement "... as high when P < 0.1 or I2 > 50%, ..."

6. Materials and Methods - Meta-analysis (page 9, line 136-139): Please check the methods applied in the study, which is my great concern, that could influence the whole results and correctness of the study. It should be that a random-effects model applys to estimate studies when their heterogeneity is high; otherwise, a fixed-effects method should be used in the case of where there is no heterogeneity between studies.

7. Materials and Methods - Meta-analysis (page 9, line 140-142): It would be better that the selected variables which used to subgroup analyses should be given with their justifications

Reviewer #3:

1. Literature search: Some Keywords or MESH terms are missed in this review that might impact the literature search results—for example, astrocytoma, oligodendroglioma, GBM, and glioblastoma multiforme.

2. Methodology: Was the grey literature considered in this review?

3. Figures 2 and 3 cannot be opened and processed. Please re-submit Figures 2 and 3 in the supported format.

4. The manuscript requires English copy-editing.

Reviewers' comments:

Reviewer's Responses to Questions

**Comments to the Author**

1. Is the manuscript technically sound, and do the data support the conclusions?

Reviewer #1: Yes

Reviewer #2: Yes

Reviewer #3: Yes

2. Has the statistical analysis been performed appropriately and rigorously? 

Reviewer #1: Yes

Reviewer #2: Yes

Reviewer #3: Yes

3. Have the authors made all data underlying the findings in their manuscript fully available?

Reviewer #1: Yes

Reviewer #2: Yes

Reviewer #3: Yes

4. Is the manuscript presented in an intelligible fashion and written in standard English?

Reviewer #1: No

Reviewer #2: Yes

Reviewer #3: No

5. Review Comments to the Author

Reviewer #1: This study represents the largest to date meta-analysis to look at the association between the XRCC3 p.Thr241Met (rs861539) polymorphism and Glioma. It is a well-designed study, which has employed appropriate statistical techniques, and its findings are interesting albeit relatively minor advance to our understanding of glioma genetic risk.

Major comment:

The authors state that the last meta study to look at rs861539 and glioma risk was over a decade ago is not true. Qi et al 2017 and Feng et al 2014 both performed similar meta-analyses but did not find there to be an association. These more recent and very similar papers should have been referenced and the authors findings discussed within the context of this early work.

Language:

Unfortunately, language quality in many parts was poor, with many sentences consisting of peculiar or inappropriate wording. Thus, I feel it is very important the authors re-write the manuscript with assistance of a copy editor.

Reviewer #2: 1. Materials and Methods - Literature search (page 8, line 111): It may be not clear to readers why the authors consider the case-control type of studies only rather than others.

2. Materials and Methods - Literature search (page 8, line 115): It is vague for the exclusion "(iii) they were duplicates(s)."

3. Materials and Methods - Extraction of data and quality appraisal (page 9, line 126): I would encourage the authors to describe more in detail about the methods and procedures of quality appraisal.

4. Materials and Methods - Meta-analysis (page 9, line 133): Please highlight and broaden the foundations and considerations of the five genetic models here or elsewhere appropriate.

5. Materials and Methods - Meta-analysis (page 9, line 137): I would suggest provide a reference for the statement "... as high when P < 0.1 or I2 > 50%, ..."

6. Materials and Methods - Meta-analysis (page 9, line 136-139): Please check the methods applied in the study, which is my great concern, that could influence the whole results and correctness of the study. It should be that a random-effects model applys to estimate studies when their heterogeneity is high; otherwise, a fixed-effects method should be used in the case of where there is no heterogeneity between studies.

7. Materials and Methods - Meta-analysis (page 9, line 140-142): It would be better that the selected variables which used to subgroup analyses should be given with their justifications.

Reviewer #3: 1. Literature search: Some Keywords or MESH terms are missed in this review that might impact the literature search results—for example, astrocytoma, oligodendroglioma, GBM, and glioblastoma multiforme.

2. Methodology: Was the grey literature considered in this review?

3. Figures 2 and 3 cannot be opened and processed. Please re-submit Figures 2 and 3 in the supported format.

4. The manuscript requires English copy-editing.

6. PLOS authors have the option to publish the peer review history of their article (what does this mean?). If published, this will include your full peer review and any attached files.

Reviewer #1: No

Reviewer #2: No

Reviewer #3: No

---

## [Author Response · Author response to Decision Letter 0]

28 Jun 2022

Academic Editor:

1. Please follow the PLOS ONE guideline for preparing the manuscript.

Response: Thank you for your comment. We have checked and made necessary changes to ensure that our manuscript follows the PLOS ONE guideline.

2. Please revise the manuscript per the Reviewers' comments.

Response: Thank you for your comment. We have revised the manuscript according to the Reviewers’ comments. The point-by-point response to reviewers is provided below.

3. It is recommended the authors mention the importance of XRCC3 polymorphism in glioma treatment. In the current practice, temozolomide is the choice chemotherapeutic for high-grade gliomas (https://www.ncbi.nlm.nih.gov/pmc/articles/PMC8651479/). Recent evidence, has noted that XRCC3 polymorphism contributes to temozolomide resistance of glioblastoma cells by mediating DNA double-strand break repair (https://pubmed.ncbi.nlm.nih.gov/29574277/). However, this may not be the whole story of XRCC3 contribution to the temozolomide resistance. Recent evidence has noted that XRCC3 contributes to mitochondrial biogenesis by facilitating the mitochondrial DNA integrity (https://pubmed.ncbi.nlm.nih.gov/29158291/). Besides, it has been shown that mitochondria improves temozolomide chemoresistance (https://www.preprints.org/manuscript/202201.0171/v2, AND https://pubmed.ncbi.nlm.nih.gov/20870728/). It is recommended the authors mention this crucial issue in the Discussion section and cite all the noted articles.

Response: Thank you for your comment. We have mentioned the issues above in our Discussion and cited the articles that you suggested (please see lines 252-260).

Reviewer #1:

This study represents the largest to date meta-analysis to look at the association between the XRCC3 p.Thr241Met (rs861539) polymorphism and Glioma. It is a well-designed study, which has employed appropriate statistical techniques, and its findings are interesting albeit relatively minor advance to our understanding of glioma genetic risk.

1. Major comment:

The authors state that the last meta study to look at rs861539 and glioma risk was over a decade ago is not true. Qi et al 2017 and Feng et al 2014 both performed similar meta-analyses but did not find there to be an association. These more recent and very similar papers should have been referenced and the authors findings discussed within the context of this early work.

Response: Thank you for your comment. We have changed the term “nearly a decade ago” to “many years ago” and cited Qi et al. 2017 and Feng et al. 2014 (in addition to the other meta-analysis in our original manuscript) (please see lines 278). Since the findings of all these meta-analyses (including Qi et al. 2017 and Feng et al. 2014) are the same, there are no other changes made to our discussion. 

2. Language:

Unfortunately, language quality in many parts was poor, with many sentences consisting of peculiar or inappropriate wording. Thus, I feel it is very important the authors re-write the manuscript with assistance of a copy editor.

Response: Thank you for your comment. We have used the service from a professional copy editor (Academic Proofreading, www.academicproofreading.uk) to revise our manuscript. We believe the language quality is now sufficient for publication. A certificate of English editing is attached.

 

Reviewer #2:

1. Materials and Methods - Literature search (page 8, line 111): It may be not clear to readers why the authors consider the case-control type of studies only rather than others.

Response: Thank you for your comment. We have added an explanation of why we considered only observational studies, such as case-control studies (please see lines 116-118).

2. Materials and Methods - Literature search (page 8, line 115): It is vague for the exclusion "(iii) they were duplicates(s)."

Response: Thank you for your comment. We have rephrased our sentence to “the study was a duplication of other publications” (please see line 120).

3. Materials and Methods - Extraction of data and quality appraisal (page 9, line 126): I would encourage the authors to describe more in detail about the methods and procedures of quality appraisal.

Response: Thank you for your comment. We have added detailed description about the methods and procedures of quality appraisal (please see lines 130-139). The quality appraisal scale has also been attached as a supplementary file.

4. Materials and Methods - Meta-analysis (page 9, line 133): Please highlight and broaden the foundations and considerations of the five genetic models here or elsewhere appropriate.

Response: Thank you for your comment. We have provided an explanation on the need to examine multiple genetic models and cited a reference which discussed the foundation of examining five genetic models (please see lines 147-149).

5. Materials and Methods - Meta-analysis (page 9, line 137): I would suggest provide a reference for the statement "... as high when P < 0.1 or I2 > 50%, ..."

Response: Thank you for your comment. We have now cited the chapter 10 of the Cochrane Handbook for Systematic Reviews of Interventions for the statement (please see line 150).

6. Materials and Methods - Meta-analysis (page 9, line 136-139): Please check the methods applied in the study, which is my great concern, that could influence the whole results and correctness of the study. It should be that a random-effects model applys to estimate studies when their heterogeneity is high; otherwise, a fixed-effects method should be used in the case of where there is no heterogeneity between studies.

Response: Thank you for your comment. It was a typographical error in our Materials and Methods. Our method was correct (we used a random-effects model when heterogeneity was high and a fixed-effects model when heterogeneity was low). This was clearly mentioned in our Results (under the heading “Quantitative data synthesis” and can also be seen in Figure 2, where the phrase “Weights are from random effects analysis” was written in the lower left corner when heterogeneity was high. We have now corrected the typographical error (please see lines 150-152).

7. Materials and Methods - Meta-analysis (page 9, line 140-142): It would be better that the selected variables which used to subgroup analyses should be given with their justifications

Response: Thank you for your comment. The selected variables are widely known to affect genetic associations. We have now added this information to our manuscript and also added a few references on this aspect (please see lines 154-155).

 

Reviewer #3:

1. Literature search: Some Keywords or MESH terms are missed in this review that might impact the literature search results—for example, astrocytoma, oligodendroglioma, GBM, and glioblastoma multiforme.

Response: Thank you for your comment. We have re-performed a search using the terms on June 10, 2022. In the first search, we obtained a different number of articles in the initial search, but after deduplication, screening by titles and abstracts, and full-text reviews, the number of included studies remained at 14. The 14 studies were exactly the same as in our original submission (please see lines 109-112 and 163-169).

2. Methodology: Was the grey literature considered in this review?

Response: Thank you for your comment. Grey literature was considered in this review, but we did not find any that fulfill the eligibility criteria. We have now added the phrase “including grey literature” in our Materials and Methods to clearly indicate that grey literature was considered in our work (please see line 108).

3. Figures 2 and 3 cannot be opened and processed. Please re-submit Figures 2 and 3 in the supported format.

Response: Thank you for your comment. All our figures are in .tif format which is supported by most publishers. We have checked our files and did not find any problem with them, and we believe the PLOS One team has also checked the files before sending them to the reviewers. However, to make it easier for you to see the figures, we have now embedded all figures into the manuscript file (please see pages 38-41).

4. The manuscript requires English copy-editing.

Response: Thank you for your comment. We have used the service from a professional copy editor (Academic Proofreading, www.academicproofreading.uk) to revise our manuscript. We believe the language quality is now sufficient for publication. A certificate of English editing is attached.

---

## [Decision Letter · Decision Letter 1]

5 Oct 2022

Association between XRCC3 p.Thr241Met polymorphism and risk of glioma: a systematic review and meta-analysis

PONE-D-21-33188R1

Dear Dr. Tan,

We’re pleased to inform you that your manuscript has been judged scientifically suitable for publication and will be formally accepted for publication once it meets all outstanding technical requirements.

Kind regards,

Yanhong Liu

Academic Editor

PLOS ONE

Additional Editor Comments (optional):

Reviewers' comments:

Reviewer's Responses to Questions

**Comments to the Author**

1. If the authors have adequately addressed your comments raised in a previous round of review and you feel that this manuscript is now acceptable for publication, you may indicate that here to bypass the “Comments to the Author” section, enter your conflict of interest statement in the “Confidential to Editor” section, and submit your "Accept" recommendation.

Reviewer #1: All comments have been addressed

Reviewer #2: All comments have been addressed

2. Is the manuscript technically sound, and do the data support the conclusions?

Reviewer #1: Yes

Reviewer #2: Yes

3. Has the statistical analysis been performed appropriately and rigorously? 

Reviewer #1: Yes

Reviewer #2: Yes

4. Have the authors made all data underlying the findings in their manuscript fully available?

Reviewer #1: Yes

Reviewer #2: Yes

5. Is the manuscript presented in an intelligible fashion and written in standard English?

Reviewer #1: Yes

Reviewer #2: Yes

6. Review Comments to the Author

Reviewer #1: (No Response)

Reviewer #2: The manuscript has been carefully and detailedly addressed in terms of my comments and I have no further questions.

7. PLOS authors have the option to publish the peer review history of their article (what does this mean?). If published, this will include your full peer review and any attached files.

Reviewer #1: No

Reviewer #2: No

---

## [Editor Report · Acceptance letter]

12 Oct 2022

PONE-D-21-33188R1 

Association between *XRCC3* p.Thr241Met polymorphism and risk of glioma: a systematic review and meta-analysis 

Dear Dr. Tan:

I'm pleased to inform you that your manuscript has been deemed suitable for publication in PLOS ONE. Congratulations! Your manuscript is now with our production department. 

Kind regards, 

on behalf of

Dr. Yanhong Liu 

Academic Editor

PLOS ONE